# Pan-Genome-Wide Association Study of Serotype 19A Pneumococci Identifies Disease-Associated Genes

Ting Li,[a] Jiayin Huang,[a] Shimin Yang,[a] Jianyu Chen,[a] Zhenjiang Yao,[a] Minghao Zhong,[b] Xinguang Zhong,[b] Xiaohua Ye[a]

[a]School of Public Health, Guangdong Pharmaceutical University, Guangzhou, China
[b]Department of Prevention and Health Care, The Sixth People's Hospital of Dongguan City, Guangdong, China

**ABSTRACT** Despite the widespread implementation of pneumococcal vaccines, hypervirulent *Streptococcus pneumoniae* serotype 19A is endemic worldwide. It is still unclear whether specific genetic elements contribute to complex pathogenicity of serotype 19A isolates. We performed a large-scale pan-genome-wide association study (pan-GWAS) of 1,292 serotype 19A isolates sampled from patients with invasive disease and asymptomatic carriers. To address the underlying disease-associated genotypes, a comprehensive analysis using three methods (Scoary, a linear mixed model, and random forest) was performed to compare disease and carriage isolates to identify genes consistently associated with disease phenotype. By using three pan-GWAS methods, we found consensus on statistically significant associations between genotypes and disease phenotypes (disease or carriage), with a subset of 30 consistently significant disease-associated genes. The results of functional annotation revealed that these disease-associated genes had diverse predicted functions, including those that participated in mobile genetic elements, antibiotic resistance, virulence, and cellular metabolism. Our findings suggest the multifactorial pathogenicity nature of this hypervirulent serotype and provide important evidence for the design of novel protein-based vaccines to prevent and control pneumococcal disease.

**IMPORTANCE** It is important to understand the genetic and pathogenic characteristics of *S. pneumoniae* serotype 19A, which may provide important information for the prevention and treatment of pneumococcal disease. This global large-sample pan-GWAS study has identified a subset of 30 consistently significant disease-associated genes that are involved in mobile genetic elements, antibiotic resistance, virulence, and cellular metabolism. These findings suggest the multifactorial pathogenicity nature of hypervirulent *S. pneumoniae* serotype 19A isolates and provide implications for the design of novel protein-based vaccines.

**KEYWORDS** *Streptococcus pneumoniae*, pneumococcus, bacterial genomics, genome-wide association study, pathogenicity, invasive pneumococcal disease, carriage

S treptococcus pneumoniae, known as the pneumococcus, is one of most important human pathogens worldwide, which asymptomatically colonizes the nasopharynx and also enters the bloodstream, leading to serious invasive pneumococcal disease (IPD). Despite the widespread implementation of a pneumococcal conjugate vaccine (PCV) immunization program, it remains the leading cause of life-threatening IPD, such as sepsis and meningitis, as well as mild noninvasive pneumococcal disease (NIPD), such as acute otitis media and sinusitis, with the highest incidence rates in the young and the elderly (1–3). Due to the increasing antibiotic resistance and disease burden, the World Health Organization has listed *S. pneumoniae* as one of the global priority pathogens. Therefore, pneumococcal disease has become a significant public health concern worldwide.

Capsular polysaccharide (CPS) is regarded as the primary virulence factor of *S. pneumoniae*, and at least 100 different serotypes have been recognized. Importantly,

Address correspondence to Xiaohua Ye, smalltomato@163.com.
The authors declare no conflict of interest.

*S. pneumoniae* serotype 19A was associated with several invasive diseases and high-level antimicrobial resistance (4–6). After the widespread use of 7-valent as well as 10-valent PCVs, serotype 19A emerged as a predominant serotype in many countries with vaccination programs and other countries without vaccination programs (7–9). In the era of 13-valent PCV (covering serotype 19A), *S. pneumoniae* 19A remains the prevalent serotype found in IPD patients (2, 10–12). Although serotype 19A pneumococci may be more prone to invade human hosts, it remains unclear whether the serotype 19A isolates from IPD patients represent a unique subgroup genetically different from those from asymptomatic individuals (11, 12). Therefore, it is important to understand the genetic and pathogenic characteristics of *S. pneumoniae* serotype 19A, which may provide important information for the prevention and treatment of *S. pneumoniae* disease and the development of proteomic vaccines in the future.

Whole-genome sequencing (WGS) with its high discriminatory power has become a feasible tool for bacterial typing, given steadily decreasing associated costs. The increasing availability of high-throughput WGS and generation of high-dimensional genomic data led to the development of numerous comparative bacterial genome analyses. Pan-genome-wide association studies (pan-GWASs) are increasingly used to explore the statistical association between genotypic variation (associated with virulence, antibiotic resistance, cellular metabolism, and disease susceptibility) and bacterial phenotypes such as disease susceptibility (13–24). To date, few studies have focused on evaluating genetic variants between pathogenic and nonpathogenic serotype 19A isolates at the pan-genome level, so it is unclear whether there are disease-associated genotypes of *S. pneumoniae*. In order to identify the key factors of *S. pneumoniae* pathogenesis, we carried out a pan-GWAS of 1,292 *S. pneumoniae* genomes (belonging to serotype 19A) available in the NCBI GenBank database. This study aimed to compare the genetic divergences between invasive disease and asymptomatic carriage isolates of *S. pneumoniae* serotype 19A so as to identify disease-associated genotypes that have a key role in the pathogenic process of *S. pneumoniae*, including the regulation of bacterial virulence, drug resistance, and transfer events of genetic elements. Our findings are essential to help understand what is causing pneumococcal disease, providing new guidance for future preventive and therapeutic interventions.

## RESULTS

**Characteristics of the *S. pneumoniae* serotype 19A isolates.** We analyzed the genomes of 1,292 *S. pneumoniae* serotype 19A isolates collected between 1905 and 2019, including 883 (68.34%) isolates from patients with invasive disease and 409 (31.66%) isolates from individuals with asymptomatic carriage. All of the IPD specimens were isolated from cerebrospinal fluid, blood, joint fluid, pleural fluid, peritoneal fluid, and so on. The numbers of the patients by age group were as follows: ≤2 years old, $n = 695$ (53.8%); 3 to 5 years old, $n = 196$ (15.2%); 6 to 59 years old, $n = 179$ (13.9%); ≥60 years old, $n = 74$ (5.7%); and age unknown, $n = 148$ (11.4%). In addition, there was a nonlinear (reverse V-shaped) annual trend for the number of the patients (Fig. 1a and b). The patients were from five continents, with the common continents being Africa ($n = 479$ [37.1%]), North America ($n = 398$ [30.8%]), and Asia ($n = 227$ [17.6%]).

**Pan-genome construction and phylogenetic analysis.** A pan-genome of the cohort consisting of 11,935 genes was constructed using 1,292 *S. pneumoniae* serotype 19A isolates. A visualization of the pangenetic results is shown in Fig. 1c to e. The diversity of the *S. pneumoniae* pan-genome, including 11,935 genes, is highlighted by the relatively large number of accessory genes ($n = 11,078$) and a high variability with a large proportion of unique genes ($n = 3,342$), suggesting that it belongs to an open pan-genome. The proportion of accessory gene increased as the sampling increased, suggesting widespread acquisition of accessory genes through horizontal gene transfer.

The most common sequence types (STs) were ST2062 ($n = 196$ [15.2%]), ST320 ($n = 174$ [13.5%]), ST199 ($n = 131$ [10.1%]), ST847 ($n = 75$ [5.8%]), ST172 ($n = 47$ [3.6%]), ST695 ($n = 44$ [3.4%]), and ST1701 ($n = 39$ [3.0%]). These STs are geographically diverse, with some geographically distant isolates separated by closely similar isolates, indicating potential geographic transmission. Despite the diversity of STs, STs clustered in

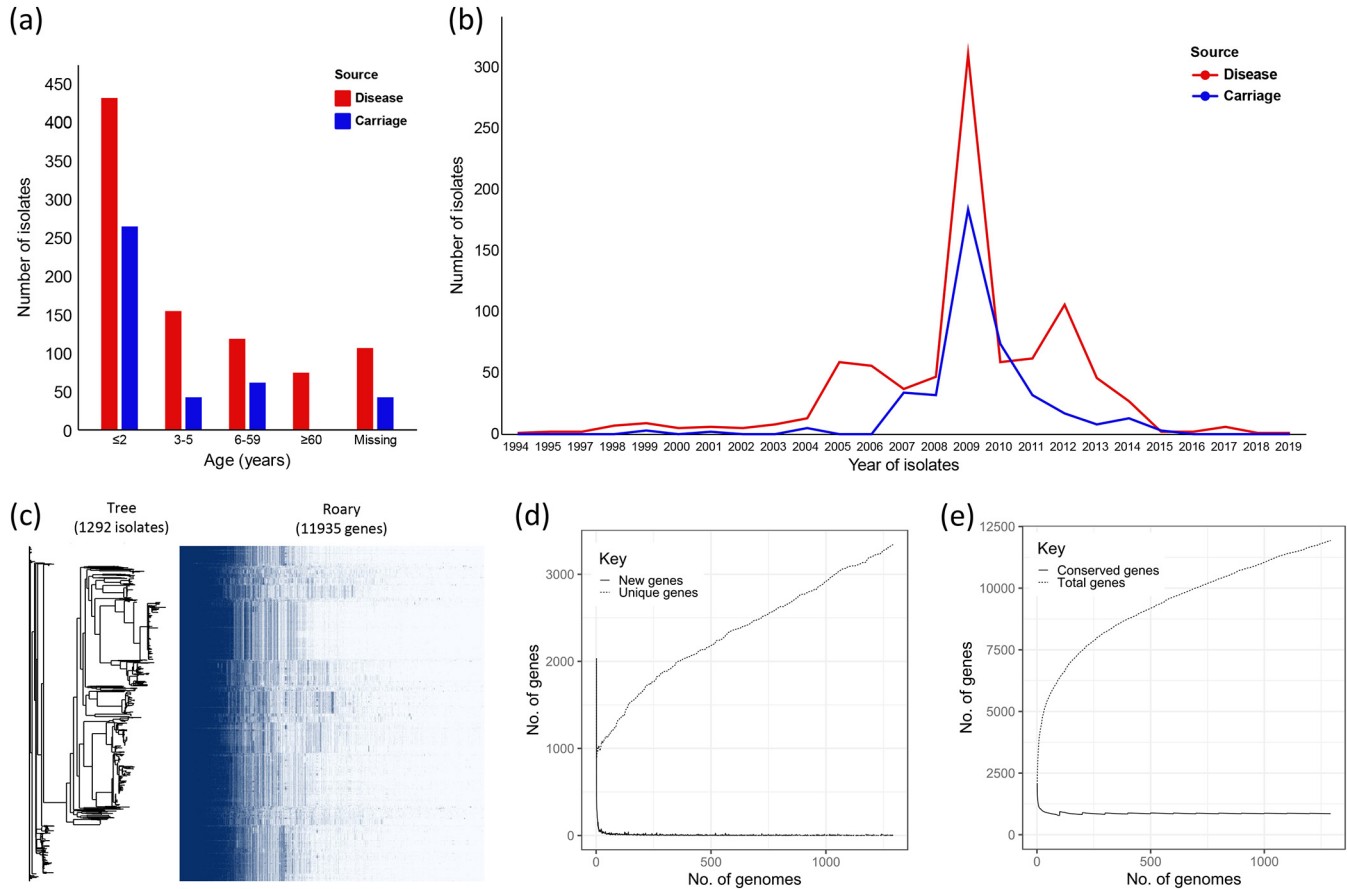

**FIG 1** Characteristics of *S. pneumoniae* serotype 19A isolates used in this study and pan-genome analysis of *S. pneumoniae* 19A genomes, constructed using Roary. (a) Distribution of pneumococcal serotype 19A isolates from disease and carriage by the age of the individuals; (b) line graph showing the distribution of sampling years of disease and carriage pneumococcal serotype 19A isolates; (c) pan-genomic visualization based on the phylogenetic tree and the matrix of genotypes (presence or absence); (d) gene accumulation curve contrasting the number of new genes and unique genes; (e) gene accumulation curve contrasting the number of total genes and conserved genes.

distinct branches of the phylogenetic tree, especially for ST2062, ST320, ST199, and ST847 (Fig. 2). Notably, comparative analysis revealed the significant differences in the proportion of specific STs between disease and carriage isolates (ST2062, 16.87% versus 11.49%, $P = 0.012$; ST320, 18.23% versus 3.18%, $P < 0.001$; ST199, 11.32% versus 7.58%, $P = 0.038$; ST847, 0.57% versus 17.11%, $P < 0.001$; ST172, 5.21% versus 0.24%, $P < 0.001$; ST695, 4.30% versus 1.47%, $P = 0.009$) (Table 1), indicating that ST2062, ST320, ST199, ST172, and ST695 isolates are associated with causing IPD.

**Multiple pan-GWAS methods for identifying disease-associated genotypes.** In the pan-GWAS analysis using the phylogenetic-based Scoary method (Fig. 3a and b), there were 828 genes statistically associated with disease phenotypes upon surviving an adjusted $P$ value threshold ($4.19 \times 10^{-6}$). The majority of disease-associated genes were annotated as hypothetical proteins, indicating that their functions were unknown. Notably, we found that ~30% of these disease-associated genes were annotated as having mobile element-related functions, indicating that horizontal gene transfer elements may be associated with causing disease.

In the pan-GWAS analysis using a linear mixed-model (LMM) method (Fig. 3c and d), there were 195 genes statistically associated with disease phenotypes upon surviving an adjusted $P$ value threshold ($1.15 \times 10^{-5}$). More than 30% of them were annotated as hypothetical proteins, and roughly 20% were annotated as mobile genetic elements.

In the pan-GWAS analysis using the random forest (RF) model, the importance of the 100 top-ranked genes is shown in Fig. 3e, with high accuracy (91.4%; 95%

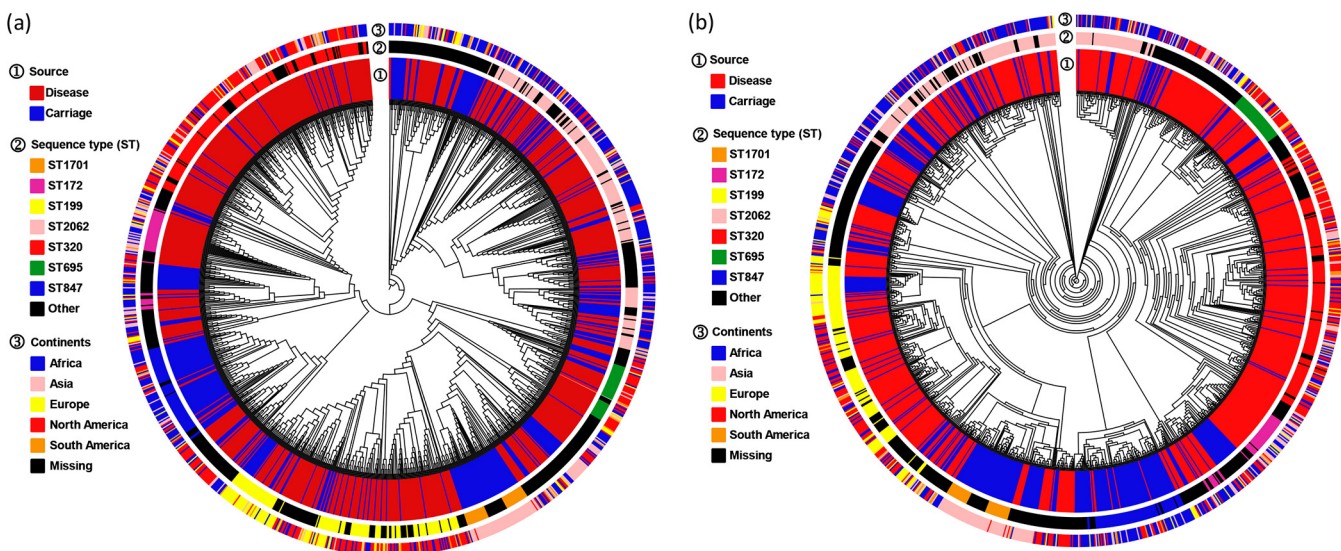

**FIG 2** Phylogenetic tree of 1,292 *S. pneumoniae* serotype 19A isolates based on core genes (a) and core SNPs (b). The colored stripes at the tips of the tree indicate isolate metadata: isolate source, sequence types, and continents.

confidence interval [CI], 89.8 to 92.9%), sensitivity (95.2%), specificity (83.1%), positive predictive value (92.4%), negative predictive value (89.0%), $\kappa$ value (0.80), and area under the concentration-time curve (AUC) value (0.89; 95% CI, 0.87 to 0.91) (Fig. 3f) as well as 10-fold cross-validation accuracy (91.2%), sensitivity (92.5%), specificity (88.7%), positive predictive value (94.9%), negative predictive value (83.1%), and $\kappa$ value (0.79).

To avoid inherent limitations of individual pan-GWAS methods, we use three methods (Scoary, LMM, and RF) to minimize false-positive associations and triangulate potential hits. Interestingly, we found consensus on statistically significant associations based on the three pan-GWAS methods, including 30 genes identified by all three methods, 86 genes identified by both Scoary and LMM, 80 genes identified by both Scoary and RF, and 36 genes identified by both RF and LMM (Fig. 4). For 30 significant genes found simultaneously by three methods, Fig. 5 illustrates the overall effect of different genes on the estimated risk score. A point above the diagonal indicates that the risk score is increased when the gene is present. Importantly, we observed 18 genes above the diagonal, suggesting that the presence of these genes may increase the risk of causing invasive disease in humans.

**Functional annotation of disease-associated genes.** After filtering the orthologous genes to identify the disease-associated genotypes in *S. pneumoniae*, there were a subset of 30 genes found simultaneously by three methods, for which 19 encoded hypothetical proteins and 11 encoded known functional proteins. These disease-associated genes had diverse predicted functions, including mobile genetic elements, antibiotic resistance, virulence, and cellular metabolism (Table 2).

The COG annotation revealed that the percentages of genes associated with (i) cell replication, recombination, and repair, (ii) cell wall, membrane, and envelope biogenesis, (iii)

**TABLE 1** Association analysis between dominant sequence types and disease phenotypes

| Sequence type | No. (%) of: | | $\chi^2$ value | P value | OR (95% CI)[a] |
|---|---|---|---|---|---|
| | Disease isolates (n = 883) | Carriage isolates (n = 409) | | | |
| ST2062 | 149 (16.87) | 47 (11.49) | 6.29 | 0.012 | 1.56 (1.10 to 2.22) |
| ST320 | 161 (18.23) | 13 (3.18) | 54.36 | <0.001 | 6.79 (3.84 to 12.02) |
| ST199 | 100 (11.32) | 31 (7.58) | 4.30 | 0.038 | 1.56 (1.02 to 2.37) |
| ST847 | 5 (0.57) | 70 (17.11) | 134.01 | <0.001 | 0.03 (0.01 to 0.08) |
| ST172 | 46 (5.21) | 1 (0.24) | 19.66 | <0.001 | 22.42 (3.80 to 906.68) |
| ST695 | 38 (4.30) | 6 (1.47) | 6.84 | 0.009 | 3.02 (1.30 to 7.03) |

[a]OR, odds ratio; 95% CI, 95% confidence interval.

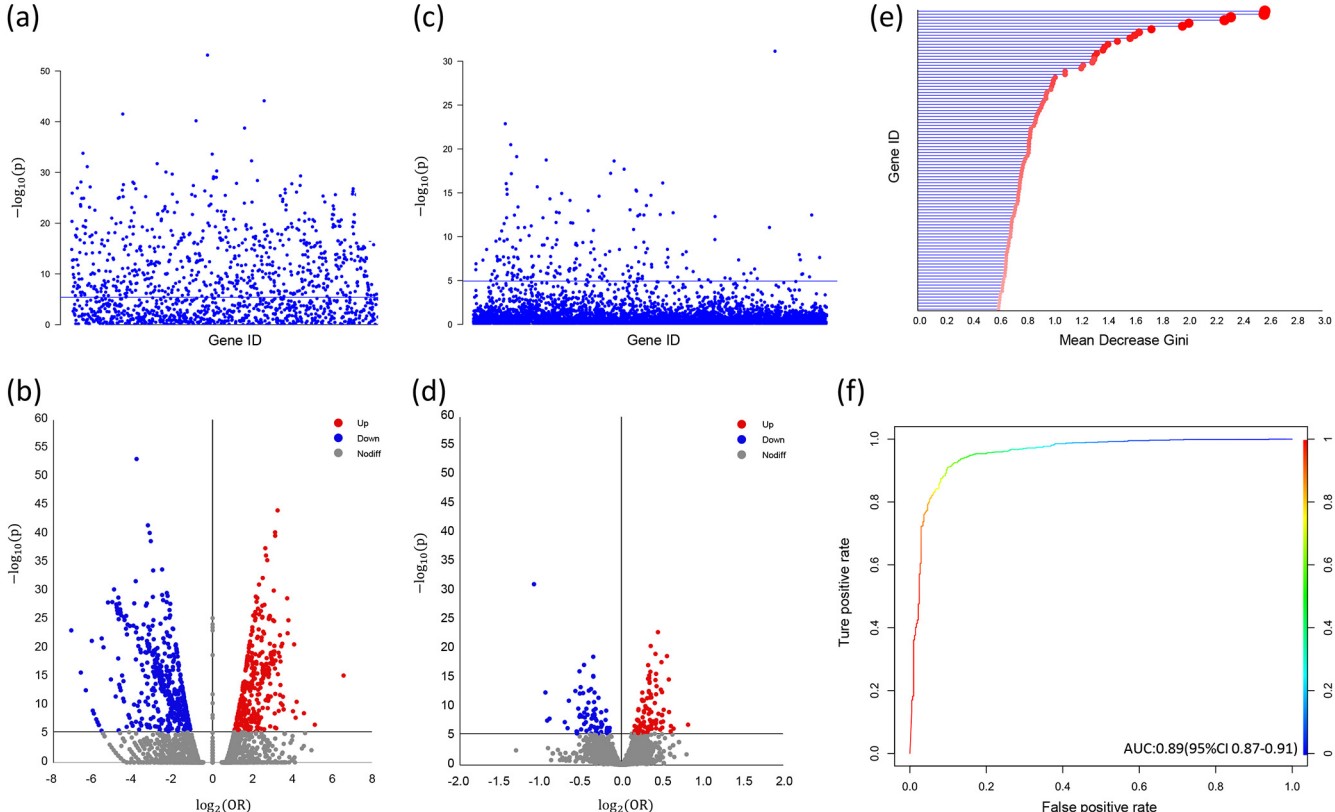

**FIG 3** pan-GWAS analyses for identifying disease-associated genotypes using the Scoary, LMM, and random forest methods. Manhattan plots show statistical significance ($-\log_{10}$) of the genes using Scoary (a) and LMM (c). Volcano plots show the relationship between statistical significance ($-\log_{10}$) and the effect size in terms of the log-transformed (base 2) carriage-to-disease odds ratio for the genetic variants using Scoary (b) and LMM (d). The horizontal lines represent the pan-WGAS statistical significance threshold based on the Bonferroni adjustment. The importance score for the 100 top-ranked genes (e) and ROC curves (f) for random forest with the 100 top-ranked genes are shown.

nucleotide transport and metabolism, and (iv) transcription were 20%, 10%, 10%, and 10%, respectively (Table 2). Furthermore, the GO annotation showed that the cytoplasm and integral component of membrane in cellular component and metal iron binding in molecular function play important roles in pathogenic process of *S. pneumoniae*.

## DISCUSSION

Despite the widespread implementation of PCV13, *S. pneumoniae* serotype 19A remains the leading cause of pneumococcal disease, especially for life-threatening IPD (1–3, 25). This phenomenon may be related to low vaccine coverage, delayed effect, antibiotic pressure, capsular switching, and immune evasion. In this study, we performed pan-GWAS analysis of 1,292 hypervirulent *S. pneumoniae* serotype 19A isolates to compare the genetic divergences between invasive disease and asymptomatic carriage isolates and found several disease-specific STs, indicating that ST2062, ST320, ST199, ST172, and ST695 isolates are associated with causing IPD. Additionally, we used three pan-GWAS methods to identify 30 simultaneously significant disease-associated genes, which had diverse predicted functions, including those participated in mobile genetic elements, antibiotic resistance, virulence and cellular metabolism. Our findings provide evidence for the presence of specific pathogenic clones and genetic elements that promote infection and invasion, suggesting the multifactorial nature of the pathogenicity.

Multilocus sequence typing in the present study showed a considerable diversity among *S. pneumoniae* 19A isolates, with the prevalent STs being ST2062, ST320, ST199, ST847, ST172, ST695, and ST1701. The *S. pneumoniae* 19A ST320 isolate, known as multidrug resistant and hypervirulent, was derived from the ancestral clone Taiwan[19F]-14

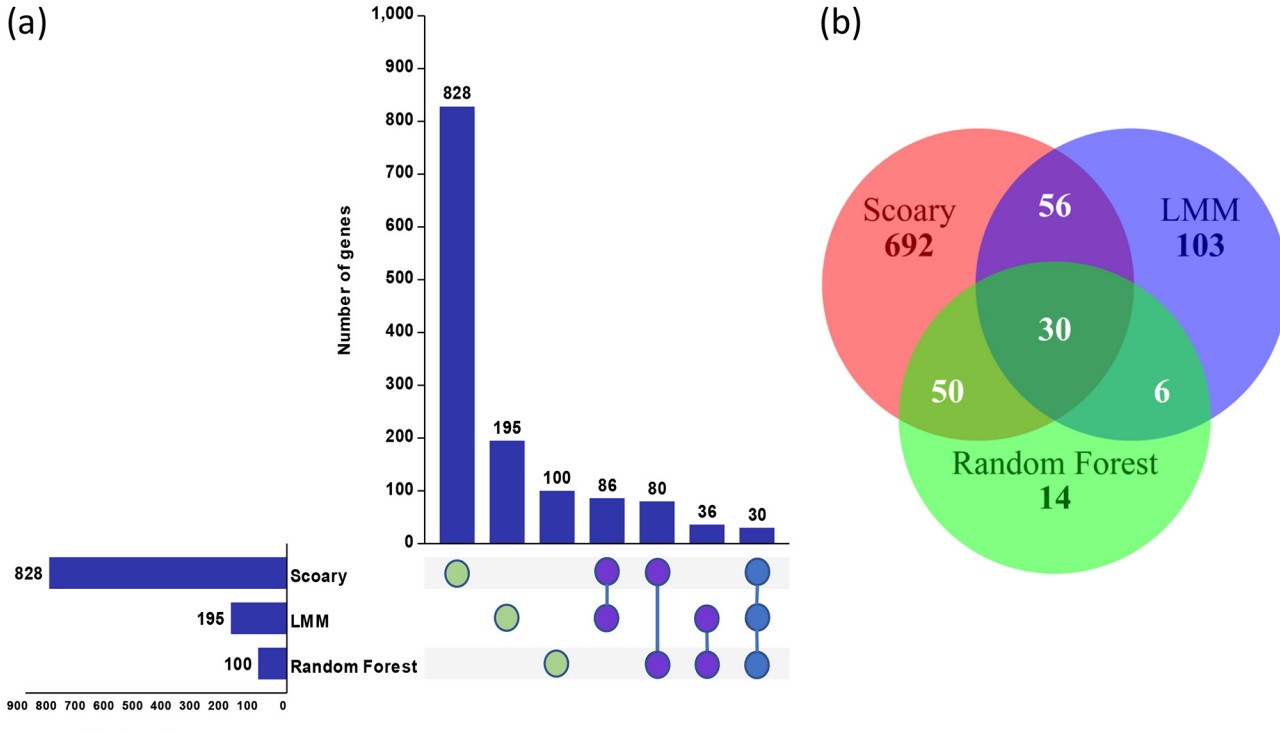

**FIG 4** Visualization of the disease-associated genes identified by three methods (Scoary, LMM, and RF) shown as intersecting sets (a) and a Venn diagram (b).

(ST236). Although ST320 is a double-locus variant of ST236, the genetic evolution from Taiwan[19F]-14 (ST236) to 19A ST320 contributed to a significant competitive advantage in enhancing colonization capacity and causing invasive disease (26). Notably, *S. pneumoniae* 19A ST320 was prevalent in many countries in both adults and children, as well as in both pre-PCV and post-PCV periods; therefore, the spread of this clone poses a threat to human health worldwide (27, 28). Despite the diversity of STs, IPD isolates were present across the phylogenetic tree, indicating that these disease clones come from multiple genetic backgrounds. Moreover, this study revealed that ST320 (odds ratio [OR] = 6.79), ST199 (OR = 1.56), ST172 (OR = 22.42), ST695 (OR = 3.02), and ST2062 (OR = 1.56) isolates were significantly and positively associated with the IPD phenotype, suggesting the present specific pathogenic clones promote invasion. Consistent with this, previous studies have demonstrated that ST320, ST199, and ST172 belonged to highly pathogenic clones and became increasing causes of infection in various geographic regions, since these STs have the ability to compete in the nasopharyngeal niche (29, 30). The above findings show a diverse genetic background of *S. pneumoniae* 19A and give an insight into the molecular typing of disease and carriage isolates.

There is increasing evidence that *S. pneumoniae* infection isolates are a subset of asymptomatic colonization clones (26), indicating that there may be specific virulence factors associated with the evolution of pneumococcal clones from harmless ancestors. In addition, the virulence factors may vary among different pneumococcal clones, so knowledge of pathogenicity determinants is crucial for identifying risk genotypes and predicting disease phenotypes during the early onset of the disease. To obtain a higher-resolution snapshot of *S. pneumoniae* genetic background, further pan-GWAS analyses are urgently needed to reveal disease-associated virulence factors, which may provide important evidence of the pathogenesis of pneumococcal disease. The latest study has revealed that the absence of consensus genetic variation associated with disease status supports the opportunistic infection model for *S. pneumoniae* serotype 1 isolates, suggesting that disease and carriage pneumococci are intrinsically hyperinvasive and

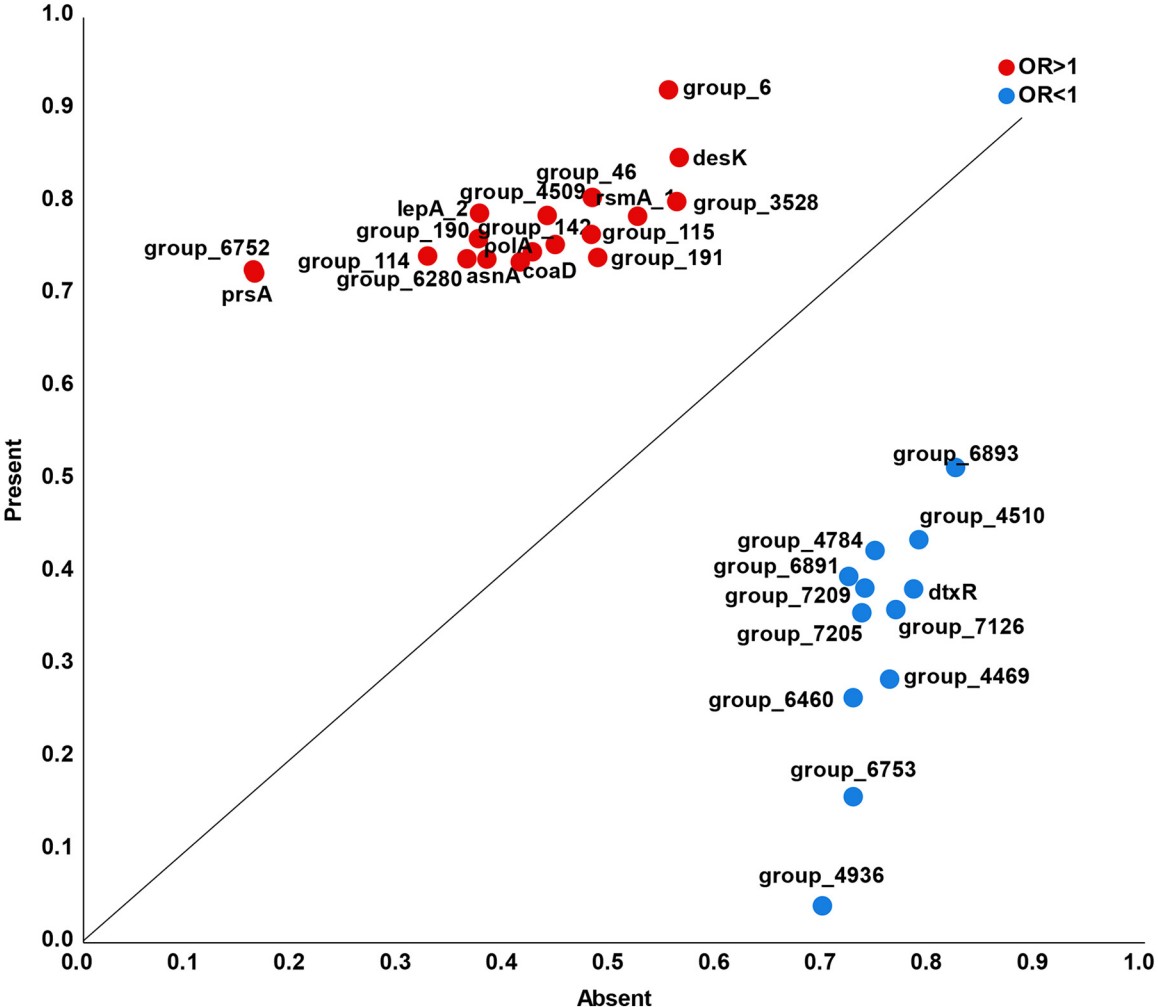

**FIG 5** Risk assessment plot for the disease-associated genes. Change in risk score for the gene when a disease-associated gene is present (*y* axis) compared to absent (*x* axis). A point above the diagonal implies that the risk score is increased when the gene is present.

equally able to cause disease (31). However, we used three pan-GWAS methods (Scoary, LMM, and RF) to triangulate potential genetic variation and surprisingly found consensus evidence that certain genetic variation is significantly overrepresented in IPD than in carriage isolates, indicating the complex multifactorial pathogenicity of *S. pneumoniae* serotype 19A. For example, the disease-associated *group_114*, *group_115*, *group_190*, and *group_142* genes were found to encode IS*5* family transposases. Transposition of genetic elements (e.g., transposons, plasmids, and some other kinds of mobile genetic elements) has significant effects on divergence and evolution of *S. pneumoniae* (32, 33). These findings have emphasized the crucial role of mobile genetic elements on the pathogenic potential of *S. pneumoniae*. In addition, *desK*, *ydhF*, and *gdhA* are involved in encoding metabolism-associated proteins such as histaminase, oxidoreductase, and NADP-specific glutamate dehydrogenase, which can modulate bacterial response to stimuli and environmental changes of *S. pneumoniae* (34, 35). Carbon and nitrogen metabolism is essential for each biological system, since every cellular component (such as proteins, genetic elements, and energy carrier molecules) is derived from metabolism. Consistent with previous studies, we observed that disease-associated *glf*, *prsA*, *dtxR*, and *rsmD* genes are involved in encoding virulence- and resistance-related proteins, which have the ability to modulate the bacterial virulence and antibiotic resistance (36–42). In summary, our findings suggest that certain genomic variations modulate *S. pneumoniae* colonization and invasion, which may shed light on the complex multifactorial nature of the pathogenicity.

**TABLE 2** Summary of the disease-associated genes identified by the pan-genome-wide association analysis

| Gene name | $P$ value | OR (95% CI)[a] | COG category[b] | Annotation |
|---|---|---|---|---|
| gdhA | $2.26 \times 10^{-6}$ | 0.84 (0.78 to 0.90) | C | NADP-specific glutamate dehydrogenase |
| coaD | $1.93 \times 10^{-14}$ | 1.24 (1.18 to 1.31) | F | Phosphopantetheine adenylyltransferase |
| asnA | $8.97 \times 10^{-15}$ | 1.25 (1.18 to 1.32) | F | Aspartate—ammonia ligase |
| group_7209 | $1.22 \times 10^{-14}$ | 0.80 (0.75 to 0.84) | F | Aspartate—ammonia ligase |
| dtxR | $9.32 \times 10^{-6}$ | 0.81 (0.74 to 0.89) | K | Diphtheria toxin repressor |
| group_6753 | $3.07 \times 10^{-7}$ | 0.80 (0.73 to 0.87) | K | Hypothetical protein |
| group_6752 | $5.42 \times 10^{-7}$ | 1.24 (1.14 to 1.35) | K | Hypothetical protein |
| group_114 | $3.07 \times 10^{-21}$ | 1.34 (1.27 to 1.43) | L | IS5 family transposase ISSpn7 |
| group_142 | $3.66 \times 10^{-11}$ | 1.18 (1.12 to 1.23) | L | IS5 family transposase ISSpn7 |
| group_115 | $3.49 \times 10^{-9}$ | 1.14 (1.09 to 1.20) | L | IS5 family transposase IS1381 |
| rsmD | $4.33 \times 10^{-12}$ | 0.81 (0.76 to 0.86) | L | Ribosomal RNA small subunit methyltransferase D |
| polA | $2.51 \times 10^{-20}$ | 1.27 (1.21 to 1.34) | L | DNA polymerase I |
| group_4784 | $2.85 \times 10^{-18}$ | 0.79 (0.76 to 0.84) | L | DNA polymerase I |
| glf[a] | $3.80 \times 10^{-7}$ | 1.30 (1.18 to 1.44) | M | UDP-galactopyranose mutase |
| group_4510 | $2.02 \times 10^{-6}$ | 0.86 (0.80 to 0.91) | M | Hypothetical protein |
| group_4509 | $9.87 \times 10^{-6}$ | 1.15 (1.08 to 1.23) | M | Hypothetical protein |
| prsA | $1.03 \times 10^{-6}$ | 1.25 (1.14 to 1.37) | O | Foldase protein PrsA |
| ydhF | $1.82 \times 10^{-16}$ | 0.74 (0.68 to 0.79) | S | Oxidoreductase YdhF |
| group_190 | $5.98 \times 10^{-16}$ | 1.24 (1.18 to 1.30) | S | IS5 family transposase ISSpn7 |
| group_4936 | $1.11 \times 10^{-31}$ | 0.43 (0.54 to 0.32) | S | Hypothetical protein |
| lepA_2 | $5.97 \times 10^{-18}$ | 1.32 (1.24 to 1.41) | S | Hypothetical protein |
| group_6280 | $2.52 \times 10^{-15}$ | 1.26 (1.19 to 1.34) | S | Hypothetical protein |
| group_4469 | $7.15 \times 10^{-12}$ | 0.79 (0.74 to 0.84) | S | Hypothetical protein |
| group_46 | $4.94 \times 10^{-10}$ | 1.22 (1.15 to 1.30) | S | Hypothetical protein |
| group_3528 | $2.22 \times 10^{-9}$ | 1.21 (1.14 to 1.28) | S | Hypothetical protein |
| group_6893 | $8.87 \times 10^{-9}$ | 0.80 (0.74 to 0.86) | S | Hypothetical protein |
| group_6891 | $7.28 \times 10^{-8}$ | 0.84 (0.79 to 0.89) | S | Hypothetical protein |
| group_191 | $1.80 \times 10^{-7}$ | 1.14 (1.09 to 1.20) | S | Hypothetical protein |
| rsmA_1 | $1.06 \times 10^{-5}$ | 1.11 (1.06 to 1.16) | S | Hypothetical protein |
| desK | $4.11 \times 10^{-11}$ | 1.32 (1.21 to 1.43) | T | Sensor histidine kinase DesK |

[a]OR, odds ratio; 95% CI, 95% confidence interval.
[b]COG annotation category: C, energy production and conversion; F, nucleotide transport and metabolism; K, transcription; L, replication, recombination, and repair; M, cell wall, membrane, and envelope biogenesis; O, posttranslational modification, protein turnover, and chaperones; S, function unknown; T, signal transduction mechanisms.

Recent advances in high-throughput WGS and its cost reductions have increased the applicability of GWAS to explore the statistical associations between genomic variation and bacterial phenotypes such as disease susceptibility and virulence, which has the potential to reveal the complex multifactorial pathogenicity and inform disease prevention measures (31). In particular, the LMM-based GWAS has led to the widespread application of genotype-phenotype association studies in bacterial genomic data, which use the random effects of kinship matrix from the phylogeny to robustly control for population structure (43). The Scoary pan-genome construction program is an easy-to-use and ultrafast tool for exploring pan-genome association between genetic variation and clinically relevant phenotypes in bacteria, which accounts for population stratification using the phylogenetic tree or Hamming distances in the genotype matrix (typically with gene presence/absence). RF, known as a nonparametric tree-based machine learning method, can effectively deal with variable interaction and correlation and also provide rapidly computable variable importance measures for variable ranking (44). This makes RF particularly attractive for the analysis of complex and high-dimensional genomic data. Many previous studies that mostly focused on single-nucleotide polymorphism (SNP)-based GWAS approaches to reveal structural variations relied on a single reference genome, but a single reference genome uncovering the complete set of genes in the entire species is inadequate for many purposes, suggesting that a pan-genome-based method is especially important for identifying disease-associated genes of the hypervirulent serotype 19A isolates (20). Considering the highly variable genomes of *S. pneumoniae* and inherent limitations of individual methods, we use three pan-GWAS methods to identify consistently significant disease-

associated genotypes, which may make our findings more comprehensive and credible to reveal the complex pathogenicity of this hypervirulent serotype.

In contrast with previous bacterial GWAS studies (28, 31, 45, 46), this study has some strengths. This study first focused on a specific highly virulent *S. pneumoniae* serotype, 19A, covering majority of countries around the world and multiple patients with a wide range of symptoms. This improved the representativeness of *S. pneumoniae* isolates and excluded the effect of capsular diversity in the pan-GWAS analysis, making our results more convincing. Additionally, we minimized the opportunity of *S. pneumoniae* serotype 19A isolates transitioning to causing IPD by including the carriage isolates from asymptomatic individuals and not infection isolates from patients with mild NIPD, which may improve the statistical power to detect disease-associated genes. Moreover, we performed pan-GWAS analyses using multiple methods to identify consistently significant genetic variants associated with IPD, thereby avoiding inherent limitations of individual methods and minimizing false-positive associations. These consensus findings may provide comprehensive evidence for clarifying the complex multifactorial pathogenicity of *S. pneumoniae* serotype 19A.

However, there are some inherent limitations that need to be considered. First, the number of disease and carriage isolates in this study was limited by current publicly available *S. pneumoniae* genomes in the NCBI GenBank database. However, this study represents a unique and large-scale genomic analysis of *S. pneumoniae* serotype 19A isolates covering a majority of countries around the world. Second, most of the disease-associated genes identified in this pan-GWAS analysis were annotated as hypothetical proteins with no known function. Based on the evidence that several hypothetical genes are consistently associated with IPD in three pan-GWAS analyses, we confirmed that these hypothetical genes are highly likely to participate in colonization and invasion. Therefore, these hypothetical proteins are worthy of further research to reveal their pathogenic mechanisms. Finally, all genes in this study were marked as having a presence or absence status, which did not take into account their extent of insertions and deletions and variation. In the future, more comprehensive SNP- and k-mer-based GWAS studies are required to confirm the effects of these disease-associated genes.

In conclusion, this global large-sample pan-GWAS study based on three methods has identified a subset of 30 consistently significant disease-associated genes that are involved in mobile genetic elements, antibiotic resistance, virulence and cellular metabolism. These findings suggest the multifactorial pathogenicity nature of hypervirulent *S. pneumoniae* serotype 19A isolates and provide important evidence for the design of novel protein-based vaccines to prevent and control pneumococcal disease.

## MATERIALS AND METHODS

**Sample selection and quality control.** A total of 1,292 genome assemblies of *S. pneumoniae* serotype 19A isolates were downloaded from the NCBI GenBank database between 1905 and 2019, which included isolates from the nasopharynx of asymptomatic carriers and clinical specimens from IPD patients. IPD was defined as an infection in which *S. pneumoniae* was recovered from normally sterile sites (e.g., blood, cerebrospinal fluid, or pleural fluid). *S. pneumoniae* assemblies were checked for low-level contamination using Kraken (v.1.1.1) (http://ccb.jhu.edu/software/kraken/) (47), which uses exact alignment to accurately assign taxonomic labels to metagenomic DNA sequences. Briefly, *S. pneumoniae* genomes were excluded from our analyses if more than 5% of the total sequence belonged to a different species. In addition, genome completeness and contamination were evaluated using the default parameters of CheckM (v.1.2.0) (48).

**Serotyping and multilocus sequence typing.** Serotyping of *S. pneumoniae* isolates were performed using SeroBA (v.1.0.2) (https://github.com/sanger-pathogens/seroba), which may infer serotypes directly from genomic data using the k-mer-based method (49). Based on the allelic profile of the seven housekeeping genes (*aroE*, *ddl*, *gdh*, *gki*, *recP*, *spi*, and *xpt*), pneumococcal sequence types (STs) were inferred by querying the pneumococcal PubMLST database (https://pubmlst.org/spneumoniae/) (50).

**Pan-genome construction and phylogenetic analysis.** The genome assemblies were reannotated using Prokka (v.1.14.6) (51). Then, these annotated assemblies were fed to Roary (v.3.13.0) for the pan-genome (core and accessory genes) construction, with the Roary parameters being 90% for minimum blastp identity and 1.5 for MCL inflation value (52). The pan-genome was visualized by the python open-source script "roary_plots.py."

A maximum likelihood phylogenetic tree based on 857 core genes was constructed with Fasttree, using the GTR+CAT model (v.2.1.11) (53). The whole-genome alignments for the variant sites with

single-nucleotide polymorphisms (SNPs) were generated by Snippy (v.4.6.0) (https://github.com/tseemann/snippy), using *S. pneumoniae* TIGR4 (NCBI accession no. PRJNA277) as the reference genome. A maximum likelihood tree based on SNPs was also generated from the recombination-filtered alignment with RAxML (v.8.2.12), using the GTR+$\Gamma$ (gamma) model and 100 bootstrap replicates (54). Visualization and annotation of the phylogenetic tree were performed using Chiplot (https://www.chiplot.online).

**pan-GWAS analysis for disease-associated genotypes.** Because pathogenicity is a complex multifactorial property, we used multiple pan-GWAS methods, including a phylogenetic-based approach (Scoary), a linear mixed model (LMM), and a machine learning method (random forest) (55) to explore the pan-GWAS between genotypes and disease phenotypes, so as to identify disease-associated genes at the pan-genome level. In the above pan-GWAS analysis, we used the disease phenotype (IPD or carriage) as the outcome variable and the pan-genome matrix of genotypes (presence or absence) as the independent variable. We carried out the univariate pan-GWAS using Scoary analysis (Scoary v.1.6.16) and LMM (Pyseer v.1.2.0) to identify disease-associated genes (56, 57), using Hamming distances and core SNPs to correct for the population structure. To control for false positives due to multiple comparisons, the Bonferroni correction was used to calculate the adjusted $P$ value threshold ($4.19 \times 10^{-6}$ for Scoary and $1.15 \times 10^{-5}$ for LMM) (58, 59). To capture the complex nonlinear association between genotypes and disease phenotypes, the RF classification model was also used to identify disease-associated genes of high variable importance, using the Random Forest package in R (v.4.1.3). The importance of the characteristic variables was rated by calculating the mean decrease in impurity (mean decrease in Gini [MDG]). We used sensitivity, specificity, positive predictive value, negative predictive value, $\kappa$ value, receiver-operating characteristic (ROC) curve, and 10-fold cross-validation to evaluate the predictive effect of the RF model. Since similar accuracy occurs when we included more genes in the RF model, the 100 top-ranked genes were regarded as a conservative threshold to reduce false-positive genes.

**Functional annotation of disease-associated genes.** All gene sequences significantly associated with disease phenotype by three methods (Scoary, LMM, and RF) were first functionally annotated by Prokka and Roary. In addition, the disease-associated gene sequences were extracted from the pan-genome using TBtools (v.1.098746), and then each DNA was checked for similarity against known genes and proteins using eggNOG (http://eggnog-mapper.embl.de/) and UniProt (https://beta.uniprot.org/).

**Data availability.** NCBI accession numbers, associated metadata, and reference for each genome included in this study are listed in the supplemental material. Detailed information on pan-GWAS analyses, DNA sequences, and annotations of genetic variants are available in the supplemental material.

## SUPPLEMENTAL MATERIAL

Supplemental material is available online only.
**SUPPLEMENTAL FILE 1**, PDF file, 0.1 MB.
**SUPPLEMENTAL FILE 2**, XLSX file, 0.2 MB.
**SUPPLEMENTAL FILE 3**, XLSX file, 0.1 MB.
**SUPPLEMENTAL FILE 4**, XLSX file, 0.02 MB.
**SUPPLEMENTAL FILE 5**, XLSX file, 0.02 MB.
**SUPPLEMENTAL FILE 6**, XLSX file, 0.02 MB.
**SUPPLEMENTAL FILE 7**, XLSX file, 0.01 MB.

## ACKNOWLEDGMENTS

This work was supported by the National Natural Science Foundation of China (no. 81973069 and 81602901), the Guangdong Basic and Applied Basic Research Foundation (no. 2019A1515010915 and 2023A1515011583), and the Key Scientific Research Foundation of Guangdong Educational Committee (no. 2022ZDZX2033). The funders had no role in the study design, data collection and analysis, and interpretation of the data.

T.L. and X.Y. designed the study and wrote the manuscript. T.L., J.H., and S.Y. performed all bioinformatic analyses. J.C. took charge of data curation. X.Y., Z.Y., M.Z., and X.Z. took charge of supervision and reviewed the data. All authors have read and approved the final manuscript.

We declare no conflict of interest.

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
