## [Reviewer comments · Microbiology Spectrum]

Microbiology Spectrum

Pan-genome-wide association study of serotype 19A pneumococci identifies disease-associated genes

Ting Li, Jiayin Huang, Shimin Yang, Jianyu Chen, Zhenjiang Yao, Minghao Zhong, Xinguang Zhong, and Xiaohua Ye

Corresponding Author(s): Xiaohua Ye, Guangdong Pharmaceutical University

Review Timeline:

Submission Date:	October 6, 2022
Editorial Decision:	February 19, 2023
Revision Received:	March 22, 2023
Accepted:	June 4, 2023

Editor: John Osei Sekyere

Reviewer(s): Disclosure of reviewer identity is with reference to reviewer comments included in decision letter(s). The following individuals involved in review of your submission have agreed to reveal their identity: Hawraa Natiq Kabroot AL-Fatlawy (Reviewer #2)

Transaction Report:

DOI: <https://doi.org/10.1128/spectrum.04073-22>

February 19, 2023

Prof. Xiaohua Ye
Guangdong Pharmaceutical University
School of Public Health
Guangdong Pharmaceutical University, 68 Nanhua Street in Chigang Road, Guangzhou, China
Guangzhou
China

Re: Spectrum04073-22 (Disease-associated genotypes of hyper-virulent serotype 19A pneumococci: a multicenter genome-wide association study)

Dear Prof. Xiaohua Ye:

Link Not Available

Sincerely,

John Osei Sekyere

Journals Department
Reviewer comments:

Reviewer #2 (Comments for the Author):

Reviewer Attachments for Manuscript Number ASM Spectrum04073-22
Disease-associated genotypes of hyper-1 virulent serotype 19A pneumococci: a multicenter genome-wide association study

There are several notes regarding your manuscript as follows.

The title

I suggest you make a few changes that would be more appropriate to the study
Keywords: it included all study subjects

Introduction

Please, add Introduction, modern reference and re-write all :

Furthermore, the World

43 Health Organization has listed *S. pneumoniae* as one of the global priority pathogens
44 due to the increasing antibiotic resistance and disease burden that pose a major threat
45 to human health. Therefore, pneumococcal disease has become a significant public health
46 concern worldwide.

47 Capsular polysaccharide (CPS) is the primary virulence factor of *S. pneumoniae*,
48 which determines at least 100 different serotypes and becomes the base of PCVs.
49 Importantly, hyper-virulent *S. pneumoniae* serotype 19A was associated with an
50 increased risk of causing invasive disease, together with the high level of antibiotic

-Discussion

add modern references to the discussion.

Conclusion :

- Need to add conclusion with number.

Conflict of Interest: need to add conflict

Bibliography/References

References

References are relevant to the study field and recent, it is recommended to use software like Mendeley to manage the bibliography list.

With best regards

Dr. Hawraa Natiq

7 / 2 / 2023

Staff Comments:

Preparing Revision Guidelines

Please return the manuscript within 60 days; if you cannot complete the modification within this time period, please contact me. If you do not wish to modify the manuscript and prefer to submit it to another journal, please notify me of your decision immediately so that the manuscript may be formally withdrawn from consideration by Microbiology Spectrum.

Reviewer Attachments for Manuscript Number ASM Spectrum04073-22

Disease-associated genotypes of hyper-1 virulent serotype 19A pneumococci: a multicenter genome-wide association study

There are several notes regarding your manuscript as follows.

The title

I suggest you make a few changes that would be more appropriate to the study

Keywords: it included all study subjects

Introduction

Please, add **Introduction**, modern reference and re-write all :

Furthermore, the World

43 Health Organization has listed *S. pneumoniae* as one of the global priority pathogens
44 due to the increasing antibiotic resistance and disease burden that post a major threat
45 to human health. Therefore, pneumococcal disease become a significant public health
46 concern worldwide.

47 Capsular polysaccharide (CPS) is the primary virulence factor of *S. pneumoniae*,
48 which determines at least 100 different serotypes and becomes the base of PCVs.

49 Importantly, hyper-virulent *S. pneumoniae* serotype 19A was associated with an
50 increased risk of causing invasive disease, together with the high level of antibiotic

-Discussion

add modern references **to the discussion.**

Conclusion :

- **Need to add conclusion with number.**

Conflict of Interest: need to add conflict

Bibliography/References

References

References are relevant to the study field and recent, it is recommended to use software like Mendeley to manage the bibliography list.

With best regards

Dr. Hawraa Natiq

7 / 2 / 2023

Mr. Editor, John Osei Sekyere Thank you very much for choosing us to review and evaluate this paper, and I hope that we will have many contributions in the future. With appreciation and respect to you

New title: Pan-genome-wide association study of serotype 19A pneumococci identifies disease-associated genes

Submission ID: Spectrum04073-22

Dear John Osei Sekyere:

Thank you for your letter and for the reviewers' comments concerning our manuscript. The comments are valuable and very helpful for revising this paper. We have studied comments carefully and have made some corrections according to the reviewer's comments. The main corrections in the paper and the responds to the reviewer's comments are as following:

Editor Comments:

Reply: Thank you for providing the opportunity to revise our manuscript. We have revised this manuscript point-by-point. We have added a doc file "Response to Reviewers" and a pdf file "Marked Up Manuscript - For Review Only" to the submitted documents.

ASM policy requires that data be available to the public upon online posting of the article, so please verify all links to sequence records, if present, and make sure that each number retrieves the full record of the data. If a new accession number is not linked or a link is broken, provide production staff with the correct URL for the record. If the accession numbers for new data are not publicly accessible before the expected

online posting of the article, publication of your article may be delayed; please contact the ASM production staff immediately with the expected release date.

Reply: We have added supplementary files. NCBI accession numbers, associated metadata, and reference for each genome included in this study are listed in the supplementary material. Detailed information for pan-GWAS analyses, DNA sequences and annotations of genetic variants are available in the supplementary material (**Lines 373-376**).

Reviewer 2 comments

The title

I suggest you make a few changes that would be more appropriate to the study.

Reply: Thank you for your suggestions. We have revised the title "Disease-associated genotypes of hyper-virulent serotype 19A pneumococci: a multicenter genome-wide association study" as "Pan-genome-wide association study of serotype 19A pneumococci identifies disease-associated genes" (**Lines 1-2**).

Keywords: it included all study subjects

Reply: We have included all study subjects in the Keyword section (**Lines 40-42**).

Introduction: Please, add Introduction, modern reference and re-write all:

Reply: Thank you for your suggestions. We have added modern references and re-write these sentences (**Lines 44-89**).

Furthermore, the World

43 Health Organization has listed *S. pneumoniae* as one of the global priority pathogens

44 due to the increasing antibiotic resistance and disease burden that pose a major threat

45 to human health. Therefore, pneumococcal disease has become a significant public health

46 concern worldwide.

Reply: Thank you. We have revised these sentences (**Lines 51-53**).

47 Capsular polysaccharide (CPS) is the primary virulence factor of *S. pneumoniae*,
48 which determines at least 100 different serotypes and becomes the base of PCVs.
49 Importantly, hyper-virulent *S. pneumoniae* serotype 19A was associated with an
50 increased risk of causing invasive disease, together with the high level of antibiotic
Reply: Thank you. We have revised these sentences (**Lines 55-58**).

Discussion

add modern references to the discussion.

Reply: Thank you for your suggestions. We have added some modern references to the discussion (**Lines 170-301**).

Conclusion:

Need to add conclusion with number.

Reply: Thank you for your suggestions. This article has only one conclusion so we have not added conclusion with number.

Conflict of Interest: need to add conflict

Reply: We have added "We declare no competing interests" in **Line 399**.

Bibliography/References

References

References are relevant to the study field and recent, it is recommended to use software like Mendeley to manage the bibliography list.

Reply: Thank you for your suggestions. We have used Mendeley to manage the bibliography list.

June 4, 2023

Prof. Xiaohua Ye
Guangdong Pharmaceutical University
School of Public Health
Guangdong Pharmaceutical University, 68 Nanhua Street in Chigang Road, Guangzhou, China
Guangzhou
China

Re: Spectrum04073-22R1 (Pan-genome-wide association study of serotype 19A pneumococci identifies disease-associated genes)

Dear Prof. Xiaohua Ye:

Your manuscript has been accepted, and I am forwarding it to the ASM Journals Department for publication. You will be notified when your proofs are ready to be viewed.

Sincerely,

John Osei Sekyere
Editor, Microbiology Spectrum
